# What Do Adolescents Learn from a 50 Minute Cardiopulmonary Resuscitation/Automated External Defibrillator Education in a Rural Area: A Pre-Post Design

**DOI:** 10.3390/ijerph16061053

**Published:** 2019-03-23

**Authors:** Ming-Fen Tsai, Li-Hsiang Wang, Ming-Shyan Lin, Mei-Yen Chen

**Affiliations:** 1Department of Nursing, Chang Gung University of Science and Technology, Taoyuan 333, Taiwan; mftsai@mail.cgust.edu.tw (M.-F.T.); lisamwang@gw.cgust.edu.tw (L.-H.W.); 2Department of Nursing, Chang Gung Memorial Hospital, Linkou 333, Taiwan; 3Department of Cardiology, Chang Gung Memorial Hospital, Chiayi 613, Taiwan; mingshyan@gmail.com; 4Department of Nursing, Chang Gung University, Taoyuan 333, Taiwan

**Keywords:** adolescents, cardiopulmonary resuscitation, automated external defibrillator, rural area

## Abstract

*Background:* Literature indicates that patients who receive cardiopulmonary resuscitation (CPR) and automated external defibrillator (AED) from bystanders have a greater chance of surviving out-of-hospital cardiac arrest (OHCA). A few evaluative studies involving CPR/AED education programs for rural adolescents have been initiated. This study aimed to examine the impact of a 50 min education program that combined CPR with AED training in two rural campuses. *Methods:* A quasi-experimental pre-post design was used. The 50 min CPR/AED training and individual performance using a Resusci Anne manikin was implemented with seventh grade students between August and December 2018. *Results:* A total of 336 participants were included in this study. The findings indicated that the 50 min CPR/AED education program significantly improved participant knowledge of emergency responses (*p* < 0.001), correct actions at home (*p* < 0.01) and outside (*p* < 0.001) during an emergency, and willingness to perform CPR if necessary (*p* < 0.001). Many participants described that “I felt more confident to perform CPR/AED,” and that “It reduces my anxiety and saves the valuable rescue time.” *Conclusions:* The brief education program significantly improved the immediate knowledge of cardiac emergency in participants and empowered them to act as first responders when they witnessed someone experiencing a cardiac arrest. Further studies should consider the study design and explore the effectiveness of such brief programs.

## 1. Introduction 

Cessation of cardiac activity outside the hospital setting, indicated by the absence of signs of circulation, is defined as out-of-hospital cardiac arrest (OHCA) [1]. Considerable evidence indicates that people who receive immediate treatments, including cardiopulmonary resuscitation (CPR) and automated external defibrillator (AED) revival, from bystanders or laypersons have a greater chance of surviving OHCA than those who do not [1,2,3,4,5,6]. Recently, mass CPR training has been considered a less expensive and easy way to increase the number of trained laypeople in a short time in many countries [5,6,7,8]. 

According to a systematic review and meta-analysis, though 53% OHCA events were witnessed by a bystander, only 32% received bystander CPR [9]. Some studies showed that people trained in CPR were more willing to perform it [10,11]. Therefore, from a public health perspective, efforts to increase survival rates should focus on the timely and effective delivery of interventions by bystanders and emergency medical service (EMS) providers. Before 2013, many people in Taiwan experienced fear or hesitation about saving people in life-threatening situations, as the law was unfavorable towards laypersons assisting in emergency. Based on the Good Samaritan Law, Article 14(2) in the Taiwan Emergency Medical Services Act [12] was amended in 2013. It was added that, with the exception of rescue personnel, the indemnification clause for emergency evaluation in the Civil Code and Criminal Code shall apply to people using emergency rescue equipment or performing first aid measures to save others from immediate life-threatening danger. Though chronic diseases contributed to seven the top 10 most common causes of death in Taiwan, accidents ranked as the sixth [13]. Therefore, in order to reduce the rate of death by accidents in recent years, the Ministry of Education and the healthcare delivery system encourages public and junior high school students to learn CPR/AED use [7].

Previous studies have shown that education programs that use AED training, use more frequent CPR training, and run for a shorter time period are associated with increased participant willingness to perform CPR [14]. In Europe, the United States, and Asia, many studies have explored the effectiveness of teaching CPR to high school students [2,5,15,16] and laypersons [14,17] via hands-on training using Resusci Anne manikins, with sessions ranging from 45 to 120 min [2,5,18]. Only a few studies have evaluated the effect of a shorter training period for junior high school students in resource-limited rural areas in Taiwan. Therefore, this study aimed to examine the impact of a 50 min CPR/AED education program in two rural campuses for junior high school students.

## 2. Materials and Methods 

### 2.1. Study Design, Sample, and Setting

This study was conducted between August and December 2018 in two middle schools with 12 classes that were located in the coastal regions of southwestern Taiwan. A one-group, pre-post test design was used. The inclusion criteria were being in the seventh grade and agreeing to participate in this study. The exclusion criterion was being absent on any of the three days of the training course and data collection. As it was difficult to match the doctors’ schedule with the training period, we invited one senior doctor and two firefighters as trainers, who were qualified, had received certified specialty training in emergency response, and used the same teaching material, e.g., PowerPoint slides, video, Resusci Anne manikins, and AED. Each class received 50 min of theoretical knowledge on topics including assessing the breath, being unconscious, cardiac arrest, and emergency and hands-on training in CPR by a doctor or a firefighter. We summarized all the lessons of basic life support for adults (Figure 1). The students were divided into five subgroups for each class. The school introduced AED in one big group at the end of the lesson.

### 2.2. Ethical Considerations

Before conducting this study, it was approved by the Institutional Review Board of Chang Gung Memorial Hospital (IRB No: 201800428B0D001). The study purpose and procedures were explained to the school’s administrators and teachers in each class.

### 2.3. Procedure and Data Collection

The 50 min session and individual practical sessions were conducted in each classroom with 24–32 students. For data collection, we used an answer sheet with 7–10 open questions. The training session in this study was developed by two of the authors. The questionnaire developed for the current study displayed good content validity (CVI = 0.92–0.98) determined by three experts (i.e., two nursing faculty members teaching in emergency nursing and one emergency doctor). Two emergency physicians agreed that the content of the questionnaire captured the important elements in the study. We also interviewed 10 sixth grade elementary students near the middle school campus to ensure that each item was clear and understandable to the youth. The teacher sent a questionnaire to students in his/her class and asked them to answer it in about 30 min before and one week after the teaching session.

### 2.4. Measurements

In the pre-test, the questions contained the following items (1 to 7), while in the post-test, items 2 to 10 were used.
(1)Have you received CPR/AED education before? When and where?(2)Please specify the emergency helpline number in Taiwan. (The answer is 119.)(3)Please specify the location of AED in the school. (The answer is based on the arrangement at each school—most of them were in the health center.)(4)How can we assess the need for CPR? (The ideal answer is tapping the shoulder to see if the patient responds; if there is no response, observe the breathing; if there is no breathing for 10 seconds, start CPR at once.)(5)What should I do at home: One day when I am at home, my grandparent suddenly falls from the chair; what should I do next?(6)What should I do on the road: When walking to school, I see that an elderly man fell on the sidewalk; what should I do?

The correct answers for items 5 and 6 are: tap the shoulder to see if the patient responds; if not, call 119 for emergency aid; meanwhile, observe if the patient on the ground is still breathing; if not, start CPR at once until the ambulance staff arrive; while calling 119, perform CPR or ask for help from others.
(7)Are you willing to perform CPR if necessary?

The post-test was conducted one week after completion of the education. Considering that many students did not receive CPR education before this education program, items 9 and 10 were added to the post-test.
(8)Please specify the compression area when performing CPR. (The ideal answer is between the two breasts or in the center of the chest ribs.)(9)Please specify the compression rates when performing the CPR per minute. (The ideal answer is 100 to 120 times per minute.)(10)What have you learnt in the last week about CPR/AED teaching?

### 2.5. Data Analysis

Data analyses were conducted using SPSS 22 (IBM SPSS, Armonk, NY: IBM Corp). The McNemar test was applied to the participants to explore their emergency knowledge and their attitude changes before and after the study. *p*-value < 0.05 was considered statistically significant.

## 3. Results 

A total of 352 participants met the inclusion criteria at the beginning of the study, 16 of which were absent on the days of the training or post-test. Finally, 336 participants completed the study. More than half were female students (56.3%), 50% had received CPR education, and nearly half of them received it one year ago (Table 1).

As shown in Table 2, the McNemar test revealed that the adolescents had significantly improved their knowledge of the emergency helpline number (*p* < 0.001) and the location of the AED in their school (*p* < 0.001). In addition, adolescents showed significant improvement in their assessment of the need for CPR (*p* < 0.001), their performance of CPR both at home (*p* < 0.001) and on the road (*p* < 0.001), and in their willingness to perform CPR if necessary (*p* < 0.001). 

As shown in Table 3, in response to what they learned after one week of the scenario-based teaching and self-practice of CPR education, many participants (51.5%) wrote “I felt more confident in using CPR and AED correctly. Finally, I can bravely perform CPR and use AED to save others in need. I am learning to be confident in using one more skill in emergency situations.” Nearly half (44.9%) reported that this training “reduced my anxiety thus saving the time that is crucial for rescuing the patient, e.g., I won’t be nervous in case a family member becomes unconscious; instead, I’ll save my beloved family member in time. If I face an emergency, I would have panicked if I had not learnt about CPR or AED. I have learnt it previously, but this time, self-practice made me feel confident.”

## 4. Discussion 

This study explored the impact of a 50 min CPR and AED education program using individual practical sessions in rural areas with limited health resources. Three key findings were highlighted in this study. First, the short CPR/AED education session significantly improved the knowledge of the adolescents and empowered them to be willing to perform CPR if necessary. Second, many adolescents reported feeling more confident and less anxious, and were prepared to make use of crucial time to rescue the patient in need of CPR/AED. Third, many students still hesitated to perform CPR after the training session.

Although half the participants had received CPR education before this program, in response to the question of compression area and compression rates per minute, only 58.3% (*n* = 196) and 53.9% (*n* = 181) were correct, respectively. Incorrect answers mostly included compression on the left chest, left or right heart, abdominal area, and “don’t know” or blank responses. Surprisingly, incorrect answers regarding the compression rates often included 30, 50, 60, 70, 80, 200, 300, 400, 500, “don’t know”, or blank responses. Why was the retention rate not adequate? Possible reasons are that we used open questions and not multiple choice questions, as did Aaberg et al. [2]. Further, adolescents may be unfamiliar with anatomical terms such as “the center of two breasts” and youth living in rural areas may find it difficult to provide the exact number of the compression rate per min on the open question. 

Considering the barriers and limitations facing the youths, further studies may use multiple choice tests with diagrams instead of descriptions, or explore innovative methods for practicum evaluation. This finding might be consistent with that of a study in Korea, by Kim et al. [18], who found that the retention rate of CPR knowledge in laypersons after 3 months of hands-on CPR training decreased, concerning the compression rate and depth. 

In addition, unlike Aaberg et al. [2], who found that 99% of Danish high school students knew how to call EMS in the case of a cardiac arrest, the present study indicated that 15.5% (*n* = 52) adolescents gave incorrect responses to questions about the emergency helpline number and 58.9% (*n* = 198) gave incorrect responses to questions about the AED location in their school before the training course. Even after the education session, 14% (*n* = 47) still incorrectly answered questions about the AED location. Some of the incorrect answers included the swimming pool (the school did not have a swimming pool), library, and student affairs room. Unlike most previous CPR studies initiated around cities and urban areas [2,4,5], the present study was conducted for adolescents living around disadvantaged areas. These circumstances suggest the need to use different teaching strategies for rural youths, such as taking them to see the location of the AED rather than just bringing it to the classroom, and making youths really dial the helpline number of 119 rather than just teaching it. Fortunately, after the educational program, respectively 95% (*n* = 319) and 90% (*n* = 302) had correct responses on what they should do at home and outside should they be the first witness.

The present study showed that adolescents reported significant increase in their willingness to perform CPR from 9.5% (*n* = 32) to 42.6% (*n* = 143) (Table 2). However, many (*n* = 193, 57.4%) were still unwilling to do so after the training session. These results are not surprising, as even CPR-trained adults commonly report that they still perceive themselves to be unable to perform CPR correctly and think that they would hurt the patient [11]. What can we do to improve willingness to perform CPR? As previous studies have shown, training sessions should start at an early age, be repeated at regular intervals, and have shorter durations, all of which might be important factors associated with the willingness of participants to perform CPR [2,14]. Further, the procedure of basic life support for adults summarized in Figure 1 might be useful for youths to remember in an emergency.

Despite the valuable findings of this study, some limitations should be noted. First, based on the absence of a control group, potential threats to the internal validity of the CPR/AED education program must be considered. Second, considering the time limitations of the instructors, we did not evaluate the quality of the educational skills of the trainers. Third, this study was conducted in the coastal regions of southwestern Taiwan, and the geographical scope may have limited the generalizability of these findings. Fourth, measurement errors may have occurred, because some youths did not respond to the open questions. 

## 5. Conclusions 

This 50 min session of CPR/AED education significantly improved knowledge and empowered adolescents to act as first responders when they witnessed someone experiencing a cardiac arrest. However, many students still hesitated to perform CPR. Further studies should consider innovative study designs for youth and examine the effectiveness of such brief programs.

## Figures and Tables

**Figure 1 ijerph-16-01053-f001:**
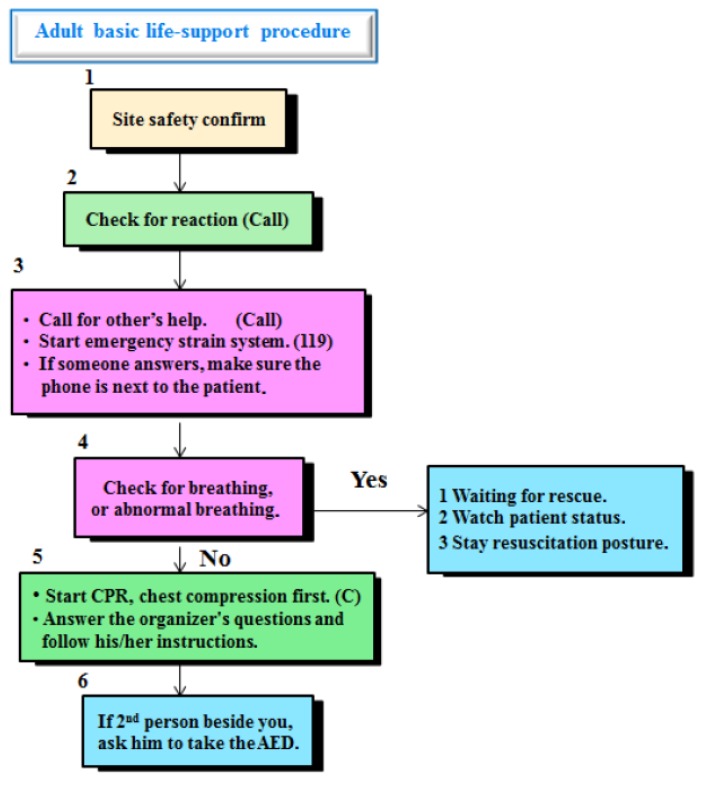
Summarized procedures for adult basic life support.

**Table 1 ijerph-16-01053-t001:** Demographic characteristics (*n* = 336).

Variables	*n* (%)	Notes	
Sex			
Male	147 (43.8)		
Female	189 (56.3)		
Had been taught CPR education			
Yes	168 (50.0)		
When and where have you learnt? Half a year ago One year ago	13 (3.9) 155 (46.1)	At school (*n* = 163, 48.5%) In the community (*n* = 5, 3%)	
No	168 (50.0)		
Compression area when doing CPR			
Incorrect	140 (41.7)	Center between two breasts (or the center of the chest between both sides of ribs)	
Correct	196 (58.3)		
Compression rate per min when doing CPR			
Incorrect	155 (46.1)	100–120 times/min	
Correct	181 (53.9)		

CPR: cardiopulmonary resuscitation.

**Table 2 ijerph-16-01053-t002:** Effects of knowledge and attitude change between pre-test and post-test (McNemar test, *n* = 336).

Variables	Pre-Test (%)	Post-Test (%)	*p*
Answer the emergency call number ^1^			<0.001
Incorrect	52 (15.5)	27 (8.0)	
Correct	284 (84.5)	309 (92.0)	
Answer the location of AED at school ^2^			<0.001
Incorrect	198 (58.9)	47 (14.0)	
Correct	138 (41.1)	289 (86.0)	
How to assess the need of CPR ^3^			<0.001
Incorrect	194 (57.7)	69 (20.5)	
Correct	142 (42.3)	267 (79.5)	
What should I do at home ^4^			<0.001
Incorrect	57 (17.0)	17 (5.1)	
Correct	279 (83.0)	319 (94.9)	
What should I do on the road ^5^			<0.001
Incorrect	102 (30.4)	34 (10.1)	
Correct	234 (69.6)	302 (89.9)	
Willingness to do CPR if necessary			<0.001
No	304 (90.5)	193 (57.4)	
Yes	32 (9.5)	143 (42.6)	

^1^ Emergency call no: 119. ^2^ Automated external defibrillator (AED) is located at school health center. ^3^ Tap the shoulder to assess his/her response, if no response, observe the breathing, if no breathing is observed in 10 seconds, start CPR at once; call 119 or asking for help are considered as correct. ^4,5^ Tap the shoulder to assess his/her response, if no response, call 119; meanwhile, observe the person on the ground to see if they are breathing; if they are not breathing, start CPR at once until the ambulance staff arrives. While calling 119, performing CPR or asking for help from others are considered as correct.

**Table 3 ijerph-16-01053-t003:** Lessons learned from the short education session and self-practice of CPR/AED.

Classification	Contents of Description	*n* (%)
I learned how to do CPR/AED and felt more confident	I felt more confident in realizing the correct procedures of CPR and AED. Performing CPR to get blood to pass into the brain is laborious. If an emergency occurs, I will do the correct CPR and use AED properly. Learning confidence in being able to accurately perform one more emergency rescuing skill. Finally, I can bravely do CPR and use AED to save others in need. Saving people by using the correct CPR procedure is sacred. Without learning accurate CPR and AED, I will lose many opportunities to save others.	173 (51.5)
I reduced my anxiety and can now take advantage of valuable rescue time	I will not be nervous if a family member falls unconscious, instead, I will save my beloved family member in time. Instant CPR and AED can snatch precious lives from death. If I was in an emergency, I would panic if I did not learn CPR or AED. I am happy to learn how to rescue the lives of others. Saving precious time by rescuing others with CPR could prevent many regretful tragedies. I have been taught previously, but this time, self-practice has made me feel confident.	151 (44.9)
I can use the accurate actions in the future	I will methodically handle such emergency cases in future. Saving people in need is true help indeed. Performing CPR is exhausting, but it can save lives in emergency. I will not just standby if people fall on the ground in future.	12 (3.6)

CPR: cardiopulmonary resuscitation; AED: automated external defibrillator.

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
