# Peer review of "What Do Adolescents Learn from a 50 Minute Cardiopulmonary Resuscitation/Automated External Defibrillator Education in a Rural Area: A Pre-Post Design"

_ijerph, 2019, doi:10.3390/ijerph16061053_

Round 1

Reviewer 1 Report

Comments and Suggestions for Authors

Abstract:  The conclusion does not appear to stem from the purpose or the methods.  The conclusion is that the program improved the participants’ immediate knowledge (not, “improved the response to the emergency situation” nor “hesitated to perform cpr” nor “responded highly inappropriately…”).   

Introduction:  It seems odd that two individual studies are referenced on lines 41-44, and then a systematic review is cited on line 45.  It would seem to me that the systematic review would be more relevant to cite and either not cite the individual studies or cite them if they are dissimilar to the systematic review’s findings.

I believe that the Introduction could be more relevant.  We already know that CPR educational programs work to improve people’s knowledge and skills of CPR immediately after the training and that the effect wears off pretty quickly.  This has been shown over and over – no need to suspect that this program will be any different.  From what I can tell, it looks like the authors think that the training can be shortened, and that it can be provided to younger students than high school students.  So, the introduction should describe what is known about the length of training programs and the appropriate age of participants.  Whether junior high school kids are the appropriate people to train for CPR is a question that is worthy of being asked.   Do young kids see many OHCAs?  I do not believe there is evidence that they do.  Therefore, no matter how short or how inexpensive the training, it is not a wise use of resources.

The purpose of the study is not stated at the end of the Introduction.  It should be clearly articulated.

Methods:   Line 89 says that the content of the questionnaire was validated by two emergency physicians.   This is not evidence of validity.  Perhaps it could be stated that 2 Eps agreed that the content of the questionnaire captured the important elements to study…

There should be some discussion of how the questionnaires were graded.   Was there interrater reliability of the grading? 

The statistical analysis should have been conducted by McNemar’s chi-square for the paired data. In addition, the impact of previous training is substantial here given that half of the class already received training within a year.  That impacts your ability to say that this program is effective because half of the students already had been trained. Without knowing how the naïve students performed relative to the previously trained students, your conclusion is suspect.

Table 2 is difficult to read with the percentages applied. Perhaps less so if just the raw numbers are provided.

Author Response

For reviewer 1:

1.      Regarding the English language and style are fine/minor spell check required.

Ans. Thank you. We have reedited the manuscript by a native English speaker.

2.      Regarding to the conclusion does not appear to stem from the purpose or the methods…

Ans. Thank you so much. We have reedited the conclusion in the abstract section. Please see the red font on p1.

3.      Regarding the introduction section, it seems odd that two individual studies are referenced on lines 41-44, and then a systematic review is cited on line 45.  It would seem to me that the systematic review would be more relevant to cite and either not cite the individual studies or cite them if they are dissimilar to the systematic review’s findings.

Ans. Thank you for this suggestion. We have reedited the introduction section on p1-2 with red font.

4.      Regarding the introduction could be more relevant…and the purpose of the study is not stated at the end of the Introduction.  It should be clearly articulated.

Ans. We have reedited the introduction section on p1-2 with red font.

5.      Regarding Line 89 that the content of the questionnaire was validated by two emergency physicians. This is not evidence of validity…and some discussion of how the questionnaires were graded…

Ans. Thank you so much. We have reedited the method section on p1-2 with red font.

6.      Regarding the statistical analysis should have been conducted by McNemar’s test for the paired data…Table 2 is difficult to read with the percentages applied.

Ans. Thank you so much for pointing out the error on this manuscript. We have reedited the manuscript and applied the McNemar’s test for the paired data. Please see the red font on p3 and table 2 on p4.

Reviewer 2 Report

This is a useful study. The English needs attention in a couple of places. However, can I suggest that all the lessons learned be summarized in a table with suggested ways to improve. It is clear that some of the students were not paying attention to the class material. The study could be extended by asking school teachers how to improve the presentation of the material.

Author Response

For reviewer 2:

Thank you for your support and encouragement that this is a useful study.

1.    Regarding the moderate English changes are required 

Ans. We have reedited the manuscript by a native English speaker from the editage company in Taiwan with the website: https://app.editage.com.tw/

2.    Regarding to some of the students was not paying attention to the class material. The study could be extended by asking school teachers how to improve the presentation of the material.

Ans. Thank you for your suggestion. We have reedited your suggestion and provided a figure with all the summarized lessons. Please see the supplementary figure 1 and manuscript on p2, p6 with red font.

Round 2

Reviewer 1 Report

The question on line 62/63 "are junior high school students suitable for training to perform CPR?" needs to be removed - it does not fit within the context of that paragraph.

The question on line 168: "Why was the retention rate not inadequate?"  needs to be rephrased to "why was the retention rate not adequate?"

Author Response

For reviewer 1:

Thank you for your comments. We have reedited the following questions.

1. Regarding the question on line 62/63 "are junior high school students suitable for training to perform CPR?" needs to be removed - it does not fit within the context of that paragraph.

Ans. Thank you so much. We have removed the above mentioned sentence accordingly.

2. Regarding the question on line 168: "Why was the retention rate not inadequate?" needs to be rephrased to "why was the retention rate not adequate?"

Ans. Thank you. We have reedited the sentence on line 168 with red font.